# A Statistical Approach to Assessing Neural Network Robustness

**Stefan Webb**[*]
Department of Engineering Science
University of Oxford

**Tom Rainforth, Yee Whye Teh**
Department of Statistics
University of Oxford

**M. Pawan Kumar**
Department of Engineering Science
University of Oxford,
Alan Turing Institute

## Abstract

We present a new approach to assessing the robustness of neural networks based on estimating the proportion of inputs for which a property is violated. Specifically, we estimate the probability of the event that the property is violated under an input model. Our approach critically varies from the formal verification framework in that when the property can be violated, it provides an informative notion of *how* robust the network is, rather than just the conventional assertion that the network is not verifiable. Furthermore, it provides an ability to scale to larger networks than formal verification approaches. Though the framework still provides a formal guarantee of satisfiability whenever it successfully finds one or more violations, these advantages do come at the cost of only providing a statistical estimate of unsatisfiability whenever no violation is found. Key to the practical success of our approach is an adaptation of multi-level splitting, a Monte Carlo approach for estimating the probability of rare events, to our statistical robustness framework. We demonstrate that our approach is able to emulate formal verification procedures on benchmark problems, while scaling to larger networks and providing reliable additional information in the form of accurate estimates of the violation probability.

## 1 Introduction

The robustness of deep neural networks must be guaranteed in mission-critical applications where their failure could have severe real-world implications. This motivates the study of neural network verification, in which one wishes to assert whether certain inputs in a given subdomain of the network might lead to important properties being violated (Zakrzewski, 2001; Bunel et al., 2018). For example, in a classification task, one might want to ensure that small perturbations of the inputs do not lead to incorrect class labels being predicted (Szegedy et al., 2013; Goodfellow et al., 2015).

The classic approach to such verification has focused on answering the binary question of whether there exist any counterexamples that violate the property of interest. We argue that this approach has two major drawbacks. Firstly, it provides no notion of how robust a network is whenever a counterexample can be found. Secondly, it creates a computational problem whenever no counterexamples exist, as formally verifying this can be very costly and does not currently scale to the size of networks used in many applications.

To give a demonstrative example, consider a neural network for classifying objects in the path of an autonomous vehicle. It will almost certainly be infeasible to train such a network that is perfectly robust to misclassification. Furthermore, because the network will most likely need to be of significant size to be effective, it is unlikely to be tractable to formally verify the network is perfectly robust, even if such a network exists. Despite this, it is still critically important to assess the robustness of the network, so that manufacturers can decide whether it is safe to deploy.

---

[*]Author for correspondence at `info@stefanwebb.me`

To address the shortfalls of the classic approach, we develop a new measure of intrinsic robustness of neural networks based on the *probability* that a property is violated under an input distribution model. Our measure is based on two key insights. The first is that for many, if not most, applications, full formal verification is neither necessary nor realistically achievable, such that one actually desires a notion of *how* robust a network is to a set of inputs, not just a binary answer as to whether it is robust or not. The second is that most practical applications have some acceptable level of risk, such that it is sufficient to show that the probability of a violation is below a certain threshold, rather than confirm that this probability is exactly zero.

By providing a probability of violation, our approach is able to address the needs of applications such as our autonomous vehicle example. If the network is not perfectly robust, it provides an explicit measure of exactly how robust the network is. If the network is perfectly robust, it is still able to tractability assert that a violation event is "probably-unsatisfiable". That is it is able to statistically conclude that the violation probability is below some tolerance threshold to true zero, even for large networks for which formal verification would not be possible.

Calculating the probability of violation is still itself a computationally challenging task, corresponding to estimating the value of an intractable integral. In particular, in most cases, violations of the target property constitute (potentially extremely) rare events. Consequently, the simple approach of constructing a direct Monte Carlo estimate by sampling from the input model and evaluating the property will be expensive and only viable when the event is relatively common. To address this, we adapt an algorithm from the Monte Carlo literature, adaptive multi-level splitting (AMLS) (Guyader et al., 2011; Nowozin, 2015), to our network verification setting. AMLS is explicitly designed for prediction of rare events and our adaptation means that we are able to reliably estimate the probability of violation, even when the true value is extremely small.

Our resulting framework is easy to implement, scales linearly in the cost of the forward operation of the neural network, and is agnostic both to the network architecture and input model. Assumptions such as piecewise linearity, Lipschitz continuity, or a specific network form are not required. Furthermore, it produces a diversity of samples which violate the property as a side-product. To summarize, our main contributions are:

- Reframing neural network verification as the estimation of the probability of a violation, thereby providing a more informative robustness metric for non-verifiable networks;

- Adaptation of the AMLS method to our verification framework to allow the tractable estimation of our metric for large networks and rare events;

- Validation of our approach on several models and datasets from the literature.

## 2 RELATED WORK

The literature on neural network robustness follows two main threads. In the optimization community, researchers seek to formally prove that a property holds for a neural network by framing it as a satisfiability problem (Zakrzewski, 2001), which we refer to as the classical approach to verification. Such methods have only been successfully scaled beyond one hidden layer networks for piecewise linear networks (Cheng et al., 2017; Katz et al., 2017), and even then these solutions do not scale to, for example, common image classification architectures with input dimensions in the hundreds, or apply to networks with nonlinear activation functions (Bunel et al., 2018). Other work has sought approximate solutions in the same general framework but still does not scale to larger networks (Pulina & Tacchella, 2010; Xiang et al., 2018; Huang et al., 2017c). As the problem is NP-hard (Katz et al., 2017), it is unlikely that an algorithm exists with runtime scaling polynomially in the number of network nodes.

In the deep learning community, research has focused on constructing and defending against adversarial attacks, and by estimating the robustness of networks to such attacks. Weng et al. (2018b) recently constructed a measure for robustness to adversarial attacks estimating a lower bound on the minimum adversarial distortion, that is the smallest perturbation required to create an adversarial example. Though the approach scales to large networks, the estimate of the lower bound is often demonstratively incorrect: it is often higher than an upper bound on the minimum adversarial distortion (Goodfellow, 2018). Other drawbacks of the method are that it cannot be applied to networks that are not Lipschitz continuous, it requires an expensive gradient computation for each class per sample, does not produce adversarial examples, and cannot be applied to non-adversarial proper-

ties. The minimum adversarial distortion is also itself a somewhat unsatisfying metric for many applications, as it conveys little information about the prevalence of adversarial examples.

In other work spanning both communities (Gehr et al., 2018; Weng et al., 2018a; Wong & Kolter, 2018), researchers have relaxed the satisfiability problem of classical verification, and are able to produce certificates-of-robustness for some samples (but not all that are robust) by giving a lower-bound on the minimal adversarial distortion. Despite these methods scaling beyond formal verification, we note that this is still a binary measure of robustness with limited informativeness.

An orthogonal track of research investigates the robustness of reinforcement learning agents to failure (Huang et al., 2017b; Lin et al., 2017). For instance, concurrent work to ours (Uesato et al., 2019) takes a continuation approach to efficiently estimating the probability that an agent fails when this may be a rare event.

## 3  MOTIVATING EXAMPLES

To help elucidate our problem setting, we consider the ACASXU dataset (Katz et al., 2017) from the formal verification literature. A neural network is trained to predict one of five correct steering decisions, such as "hard left," "soft left," etc., for an unmanned aircraft to avoid collision with a second aircraft. The inputs $\mathbf{x}$ describe the positions, orientations, velocities, etc. of the two aircraft. Ten interpretable properties are specified along with corresponding constraints on the inputs, for which violations correspond to events causing collisions. Each of these properties is encoded in a function, $s$, such that it is violated when $s(\mathbf{x}) \geq 0$. The formal verification problem asks the question, "Does there exist an input $\mathbf{x} \in \mathcal{E} \subseteq \mathcal{X}$ in a constrained subset, $\mathcal{E}$, of the domain such that the property is violated?" If there exists a counterexample violating the property, we say that the property is satisfiable (SAT), and otherwise, unsatisfiable (UNSAT).

Another example is provided by adversarial properties from the deep learning literature on datasets such as MNIST. Consider a neural network $f_\theta(\mathbf{x}) = \mathrm{Softmax}(\mathbf{z}(\mathbf{x}))$ that classifies images, $\mathbf{x}$, into $C$ classes, where the output of $f$ gives the probability of each class. Let $\delta$ be a small perturbation in an $l_p$-ball of radius $\epsilon$, that is, $\|\delta\|_p < \epsilon$. Then $\mathbf{x} = \mathbf{x}' + \delta$ is an adversarial example for $\mathbf{x}'$ if $\arg\max_i \mathbf{z}(\mathbf{x})_i \neq \arg\max_i \mathbf{z}(\mathbf{x}')_i$, i.e. the perturbation changes the prediction. Here, the property function is $s(\mathbf{x}) = \max_{i \neq c}(\mathbf{z}(\mathbf{x})_i - \mathbf{z}(\mathbf{x})_c)$, where $c = \arg\max_j \mathbf{z}(\mathbf{x}')_j$ and $s(\mathbf{x}) \geq 0$ indicates that $\mathbf{x}$ is an adversarial example. Our approach subsumes adversarial properties as a specific case.

## 4  ROBUSTNESS METRIC

The framework for our robustness metric is very general, requiring only a) a neural network $f_\theta$, b) a property function $s(\mathbf{x}; f, \phi)$, and c) an input model $p(\mathbf{x})$. Together these define an integration problem, with the main practical challenge being the estimation of this integral. Consequently, the method can be used for any neural network. The only requirement is that we can evaluate the property function, which typically involves a forward pass of the neural network.

The property function, $s(\mathbf{x}; f_\theta, \phi)$, is a deterministic function of the input $\mathbf{x}$, the trained network $f_\theta$, and problem specific parameters $\phi$. For instance, in the MNIST example, $\phi = \arg\max_i f_\theta(\mathbf{x}')_i$ is the true output of the unperturbed input. Informally, the property reflects how badly the network is performing with respect to a particular property. More precisely, the event

$$E \triangleq \{s(\mathbf{x}; f_\theta, \phi) \geq 0\} \tag{1}$$

represents the property being violated. Predicting the occurrence of these, typically rare, events will be the focus of our work. We will omit the dependency on $f_\theta$ and $\phi$ from here on for notional conciseness, noting that these are assumed to be fixed and known for verification problems.

The input model, $p(\mathbf{x})$, is a distribution over the subset of the input domain that we are considering for counterexamples. For instance, for the MNIST example we could use $p(\mathbf{x}; \mathbf{x}') \propto \mathbb{1}(\|\mathbf{x} - \mathbf{x}'\|_p \leq \epsilon)$ to consider uniform perturbations to the input around an $l_p$-norm ball with radius $\epsilon$. More generally, the input model can be used to place restrictions on the input domain and potentially also to reflect that certain violations might be more damaging than others.

Together, the property function and input model specify the probability of failure through the integral

$$\mathcal{I}[p, s] \triangleq P_{X \sim p(\cdot)}(s(X) \geq 0) = \int_{\mathcal{X}} \mathbb{1}_{\{s(\mathbf{x}) \geq 0\}} p(\mathbf{x}) d\mathbf{x}. \tag{2}$$

This integral forms our measure of robustness. The integral being equal to exactly zero corresponds to the classical notion of a formally verifiable network. Critically though, it also provides a measure for *how* robust a non-formally-verifiable network is.

## 5 METRIC ESTIMATION

Our primary goal is to estimate (2) in order to obtain a measure of robustness. Ideally, we also wish to generate example inputs which violate the property. Unfortunately, the event $E$ is typically very rare in verification scenarios. Consequently, the estimating the integral directly using Monte Carlo,

$$\hat{P}_{X\sim p(\cdot)}\left(s(X) \geq 0\right) = \frac{1}{N}\sum_{n=1}^{N} \mathbb{1}_{\{s(\mathbf{x}_n)\geq 0\}}, \quad \text{where} \quad \mathbf{x}_n \stackrel{\text{i.i.d.}}{\sim} p(\cdot) \tag{3}$$

is typically not feasible for real problems, requiring an impractically large number of samples to achieve a reasonable accuracy. Even when $E$ is not a rare event, we desire to estimate the probability using as few forward passes of the neural network as possible to reduce computation. Furthermore, the dimensionality of $\mathbf{x}$ is typically large for practical problems, such that it is essential to employ a method that scales well in dimensionality. Consequently many of the methods commonly employed for such problems, such as the cross-entropy method (Rubinstein, 1997; De Boer et al., 2005), are inappropriate due to relying on importance sampling, which is well known to scale poorly.

As we will demonstrate empirically, a less well known but highly effective method from the statistics literature, adaptive multi-level splitting (AMLS) (Kahn & Harris, 1951; Guyader et al., 2011), can be readily adapted to address all the aforementioned computational challenges. Specifically, AMLS is explicitly designed for estimating the probability of rare events and our adaptation is able to give highly accurate estimates even when the $E$ is very rare. Furthermore, as will be explained later, AMLS also allows the use of MCMC transitions, meaning that our approach is able to scale effectively in the dimensionality of $\mathbf{x}$. A further desirable property of AMLS is that it produces property-violating examples as a side product, namely, it produces samples from the distribution

$$\pi(\mathbf{x}) \triangleq p(\mathbf{x} \mid E) = p(\mathbf{x})\mathbb{1}_{\{s(\mathbf{x})\geq 0\}}/\mathcal{I}\left[p, s\right]. \tag{4}$$

Such samples could, in theory, be used to perform robust learning, in a similar spirit to Goodfellow et al. (2015) and Madry et al. (2017).

### 5.1 MULTI-LEVEL SPLITTING

Multi-level splitting (Kahn & Harris, 1951) divides the problem of predicting the probability of a rare event into several simpler ones. Specifically, we construct a sequence of intermediate targets,

$$\pi_k(\mathbf{x}) \triangleq p(\mathbf{x} \mid \{s(\mathbf{x}) \geq L_k\}) \propto p(\mathbf{x})\mathbb{1}_{\{s(\mathbf{x})\geq L_k\}}, \quad k = 0, 1, 2, \dots, K,$$

for levels, $-\infty = L_0 < L_1 < L_2 < \cdots < L_K = 0$, to bridge the gap between the input model $p(\mathbf{x})$ and the target $\pi(\mathbf{x})$. For any choice of the intermediate levels, we can now represent equation (2) through the following factorization,

$$P_{X\sim p(\cdot)}\left(s(X) \geq 0\right) = \prod_{k=1}^{K} P\left(s(X) \geq L_k \mid s(X) \geq L_{k-1}\right) = \prod_{k=1}^{K} P_k,$$

$$\text{where} \quad P_k \triangleq \mathbb{E}_{X\sim \pi_{k-1}(\cdot)}\left[\mathbb{1}_{\{s(X)\geq L_k\}}\right]. \tag{5}$$

Provided consecutive levels are sufficiently close, we will be able to reliably estimate each $P_k$ by making use of the samples from one level to initialize the estimation of the next.

Our approach starts by first drawing $N$ samples, $\{\mathbf{x}_n^{(0)}\}_{n=1}^{N}$, from $\pi_0(\cdot) = p(\cdot)$, noting that this can be done exactly because the perturbation model is known. These samples can then be used to estimate $P_1$ using simple Monte Carlo,

$$P_1 \approx \hat{P}_1 \triangleq \frac{1}{N}\sum_{n=1}^{N} \mathbb{1}_{\{s(\mathbf{x}_n^{(0)})\geq L_1\}} \quad \text{where} \quad \mathbf{x}_n^{(0)} \sim \pi_0(\cdot).$$

In other words, $P_1$ is the fraction of these samples whose property is greater than $L_1$. Critically, by ensuring the value of $L_1$ is sufficiently small for $\{s(\mathbf{x}_n) \geq L_1\}$ to be a common event, we can ensure $\hat{P}_1$ is a reliable estimate for moderate numbers of samples $N$.

---

**Algorithm 1** Adaptive multi-level splitting with termination criterion

---

1: **Input:** Input model $p(\mathbf{x})$, sample quantile $\rho$, MH proposal $g(\mathbf{x}'|\mathbf{x})$, number of MH steps $M$, termination threshold $\log(P_{\min})$
2: Sample $\{\mathbf{x}_n^{(0)}\}_{n=1}^N$ i.i.d. from $p(\cdot)$
3: Initialize $L \leftarrow -\infty, \quad L_{\text{prev}} \leftarrow -\infty, \quad \log(\mathcal{I}) \leftarrow 0, \quad k \leftarrow 0$
4: **while** $L < 0$ **do**
5:     $k \leftarrow k + 1$
6:     Evaluate and sort $\{s(\mathbf{x}_n^{(k-1)})\}_{n=1}^N$ in descending order
7:     $L_k \leftarrow \min\{0, s(\mathbf{x}_{\lfloor \rho N \rfloor}^{(k-1)})\}$                                       ▷ Updating the level
8:     $\hat{P}_k \leftarrow \#\{\mathbf{x}_n^{(k-1)} \mid s(\mathbf{x}_n^{(k-1)}) \geq L\} / N$
9:     $\log(\mathcal{I}) \leftarrow \log(\mathcal{I}) + \log(\hat{P}_k)$                          ▷ Updating integral estimate
10:     **if** $\log(\mathcal{I}) < \log(P_{\min})$ **then return** $(\emptyset, -\infty)$ **end if**     ▷ Final estimate will be less than $\log(P_{\min})$
11:     Initialize $\{\mathbf{x}_n^{(k)}\}_{n=1}^N$ by resampling with replacement $N$ times from $\{\mathbf{x}_n^{(k-1)} \mid s(\mathbf{x}_n^{(k-1)}) \geq L\}$
12:     Apply $M$ MH updates separately to each $\mathbf{x}_n^{(k)}$ using $g(\mathbf{x}'|\mathbf{x})$
13:     [Optional] Adapt $g(\mathbf{x}'|\mathbf{x})$ based on MH acceptance rates
14: **end while**
15: **return** $(\{\mathbf{x}_n^{(k)}\}_{n=1}^N, \log(\mathcal{I}))$

---

To estimate the other $P_k$, we need to be able to draw samples from $\pi_{k-1}(\cdot)$. For this we note that if $\{\mathbf{x}_n^{(k-2)}\}_{n=1}^N$ are distributed according to $\pi_{k-2}(\cdot)$, then the subset of these samples for which $s(\mathbf{x}_n^{(k-2)}) \geq L_{k-1}$ are distributed according to $\pi_{k-1}(\cdot)$. Furthermore, setting $L_{k-1}$ up to ensure this event is not rare means a significant proportion of the samples will satisfy this property.

To avoid our set of samples shrinking from one level to the next, it is necessary to carry out a rejuvenation step to convert this smaller set of starting samples to a full set of size $N$ for the next level. To do this, we first carry out a uniform resampling with replacement from the set of samples satisfying $s(\mathbf{x}_n^{(k-1)}) \geq L_k$ to generate a new set of $N$ samples which are distributed according to $\pi_k(\cdot)$, but with a large number of duplicated samples. Starting with these samples, we then successively apply $M$ Metropolis–Hastings (MH) transitions targeting $\pi_k(\cdot)$ separately to each sample to produce a fresh new set of samples $\{\mathbf{x}_n^{(k)}\}_{n=1}^N$ (see Appendix A for full details). These samples can then in turn be used to form a Monte Carlo estimate for $P_k$,

$$P_k \approx \hat{P}_k \triangleq \frac{1}{N} \sum_{n=1}^N \mathbb{1}_{\{s(\mathbf{x}_n^{(k-1)}) \geq L_k\}} \quad \text{where} \quad \mathbf{x}_n^{(k-1)} \sim \pi_{k-1}(\cdot), \tag{6}$$

along with providing the initializations for the next level. Running more MH transitions decreases the correlations between the set of samples, improving the performance of the estimator.

We have thus far omitted to discuss how the levels $L_k$ are set, other than asserting the need for the levels to be sufficiently close to allow reliable estimation of each $P_k$. Presuming that we are also free to choose the number of levels $K$, there is inevitably a trade-off between ensuring that each $\{s(X) \geq L_k\}$ is not rare given $\{s(X) \geq L_{k-1}\}$, and keeping the number of levels small to reduce computational costs and avoid the build-up of errors. AMLS (Guyader et al., 2011) builds on the basic multi-level splitting process, providing an elegant way of controlling this trade-off by adaptively selecting the level to be the minimum of 0 and some quantile of the property under the current samples. The approach terminates when the level reaches zero, such that $L_K = 0$ and $K$ is a dynamic parameter chosen implicitly by the adaptive process.

Choosing the $\rho$th quantile of the values of the property results in discarding a fraction $(1 - \rho)$ of the chains at each step of the algorithm. This allows explicit control of the rarity of the events to keep them at a manageable level. We note that if all the sample property values are distinct, then this approach gives $P_k = \rho, \forall k < K$. To give intuition to this, we can think about splitting up $\log(\mathcal{I})$ into chunks of size $\log(\rho)$. For any value of $\log(\mathcal{I})$, there is always a unique pair of values $\{K, P_K\}$ such that $\log(\mathcal{I}) = K \log(\rho) + \log(P_K)$, $K \geq 0$ and $P_K < \rho$. Therefore the problem of estimating $\mathcal{I}$ is equivalent to that of estimating $K$ and $P_K$.

## 5.2 TERMINATION CRITERION

The application of AMLS to our verification problem presents a significant complicating factor in that the true probability of our rare event might be exactly zero. Whenever this is the case, the basic AMLS approach outlined in (Guyader et al., 2011) will never terminate as the quantile of the property will never rise above zero; the algorithm simply produces closer and closer intermediate levels as it waits for the event to occur.

To deal with this, we introduce a termination criterion based on the observation that AMLS's running estimate for $\mathcal{I}$ monotonically decreases during running. Namely, we introduce a threshold probability, $P_{\min}$, below which the estimates will be treated as being numerically zero. We then terminate the algorithm if $\mathcal{I} < P_{\min}$ and return $\mathcal{I} = 0$, safe in the knowledge that even if the algorithm would eventually generate a finite estimate for $\mathcal{I}$, this estimate is guaranteed to be less than $P_{\min}$.

Putting everything together, gives the complete method as shown in Algorithm 1. See Appendix B for low-level implementation details.

## 6 EXPERIMENTS

### 6.1 EMULATION OF FORMAL VERIFICATION

In our first experiment[1], we aim to test whether our robustness estimation framework is able to effectively emulate formal verification approaches, while providing additional robustness information for SAT properties. In particular, we want to test whether it reliably identifies properties as being UNSAT, for which $\mathcal{I} = 0$, or SAT, for which $\mathcal{I} > 0$. We note that the method still provides a formal demonstration for SAT properties because having a non-zero estimate for $\mathcal{I}$ indicates that at least one counterexample has been found. Critically, it further provides a measure for how robust SAT properties are, through its estimate for $\mathcal{I}$.

We used the COLLISIONDETECTION dataset introduced in the formal verification literature by (Ehlers, 2017). It consists of a neural network with six inputs that has been trained to classify two car trajectories as colliding or non-colliding. The architecture has 40 linear nodes in the first layer, followed by a layer of max pooling, a ReLU layer with 19 hidden units, and an output layer with 2 hidden units. Along with the dataset, 500 properties are specified for verification, of which 172 are SAT and 328 UNSAT. This dataset was chosen because the model is small enough so that the properties can be formally verified. These formal verification methods do not calculate the value of $\mathcal{I}$, but rather confirm the existence of a counterexample for which $s(\mathbf{x}) > 0$.

We ran our approach on all 500 properties, setting $\rho = 0.1$, $N = 10^4$, $M = 1000$ (the choice of these hyperparameters will be justified in the next subsection), and using a uniform distribution over the input constraints as the perturbation model, along with a uniform random walk proposal. We compared our metric estimation approach against the naive Monte Carlo estimate using $10^{10}$ samples. The generated estimates of $\mathcal{I}$ for all SAT properties are shown in Figure 1a.

Both our approach and the naive MC baseline correctly identified all of the UNSAT properties by estimating $\mathcal{I}$ as exactly zero. However, despite using substantially more samples, naive MC failed to find a counterexample for 8 of the rarest SAT properties, thereby identifying them as UNSAT, whereas our approach found a counterexample for all the SAT properties. As shown in Figure 1a, the variances in the estimates for $\mathcal{I}$ of our approach were also very low and matched the unbiased MC baseline estimates for the more commonly violated properties, for which the latter approach still gives reliable, albeit less efficient, estimates. Along with the improved ability to predict rare events, our approach was also significantly faster than naive MC throughout, with a speed up of several orders of magnitude for properties where the event is not rare—a single run with naive MC took about 3 minutes, whereas a typical run of ours took around 3 seconds.

### 6.2 SENSITIVITY TO PARAMETER SETTINGS

As demonstrated by Bréhier et al. (2015), AMLS is unbiased under the assumption that perfect sampling from the targets, $\{\pi_k\}_{k=1}^{K-1}$, is possible, and that the cumulative distribution function of $s(X)$ is continuous. In practice, finite mixing rates of the Markov chains and the dependence between the

---

[1]Code to reproduce all experimental results is available at `https://github.com/oval-group/statistical-robustness`.

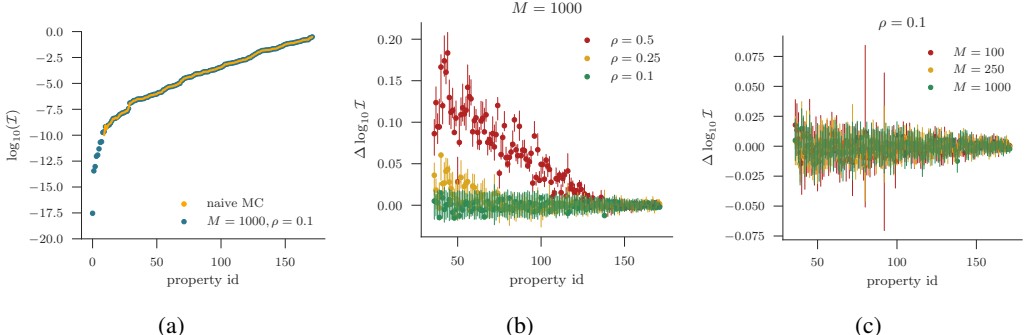

(a)                                   (b)                                   (c)

Figure 1: (a) Estimate of $\mathcal{I}$ for all SAT properties of COLLISIONDETECTION problem. Error bars indicating $\pm$ three standard errors from 30 runs are included here and throughout, but the variance of the estimates was so small that these are barely visible. We can further conclude low bias of our method for the properties where naive MC estimation was feasible, due to the fact that naive MC produces unbiased (but potentially high variance) estimates. (b) Mean AMLS estimate relative to naive MC estimate for different $\rho$ holding $M = 1000$ fixed, for those properties with $\log_{10} \mathcal{I} > -6.5$ such that they could be estimated accurately. The bias decreases both as $\rho$ and the rareness of the event decrease. (c) As per (b) but with varying $M$ and holding $\rho = 0.1$ fixed.

initialization points for each target means that sampling is less than perfect, but improves with larger values of $M$ and $N$. The variance, on the other hand, theoretically strictly decreases with larger values of $N$ and $\rho$ (Bréhier et al., 2015).

In practice, we found that while larger values of $M$ and $N$ were always beneficial, setting $\rho$ too high introduced biases into the estimate, with $\rho = 0.1$ empirically providing a good trade-off between bias and variance. Furthermore, this provides faster run times than large values of $\rho$, noting that the smaller values of $\rho$ lead to larger gaps in the levels.

To investigate the effect of the parameters more formally, we further ran AMLS on the SAT properties of COLLISIONDETECTION, varying $\rho \in \{0.1, 0.25, 0.5\}$, $N \in \{10^3, 10^4, 10^5\}$ and $M \in \{100, 250, 1000\}$, again comparing to the naive MC estimate for $10^{10}$ samples. We found that the value of $N$ did not make a discernible difference in this range regardless of the values for $\rho$ and $M$, and thus all presented results correspond to setting $N = 10^4$. As shown in Figure 1b, we found that the setting of $\rho$ made a noticeable difference to the estimates for the relatively rarer events. All the same, these differences were small relative to the differences between properties. As shown in Figure 1c, the value of $M$ made little difference when $\rho = 0.1$,. Interesting though, we found that the value of $M$ was important for different values of $\rho$, as shown in Appendix C.1, with larger values of $M$ giving better results as expected.

## 6.3 CONVERGENCE WITH HIGHER-DIMENSIONAL INPUTS AND LARGER NETWORKS

To validate the algorithm on a higher-dimensional problem, we first tested adversarial properties on the MNIST and CIFAR−10 datasets using a dense ReLU network with two hidden-layer of size 256. An $l_\infty$-norm ball perturbation around the data point with width $\epsilon$ was used as the uniform input model, with $\epsilon = 1$ representing an $l_\infty$-ball filling the entire space (the pixels are scaled to $[0, 1]$), together with a uniform random walk MH proposal. After training the classifiers, multilevel splitting was run on ten samples from the test set at multiple values of $\epsilon$, with $N = 10000$ and $\rho = 0.1$, and $M \in \{100, 250, 1000\}$ for MNIST and $M \in \{100, 250, 500, 1000, 2000\}$ for CIFAR−10. The results for naive MC were also evaluated using $5 \times 10^9$ samples—less than the previous experiment as the larger network made estimation more expensive—in the cases where the event was not too rare. This took around twenty minutes per naive MC estimate, versus a few minutes for each AMLS estimate.

As the results were similar across datapoints, we present the result for a single example in the top two rows of Figure 2. As desired, a smooth curve is traced out as $\epsilon$ decreases, for which the event $E$ becomes rarer. For MNIST, acceptable accuracy is obtained for $M = 250$ and high accuracy results for $M = 1000$. For CIFAR−10, which has about four times the input dimension of MNIST, larger values of $M$ were required to achieve comparable accuracy. The magnitude of $\epsilon$ required to give a certain value of $\log(\mathcal{I})$ is smaller for CIFAR−10 than MNIST, reflecting that adversarial examples for the former are typically more perceptually similar to the datapoint.

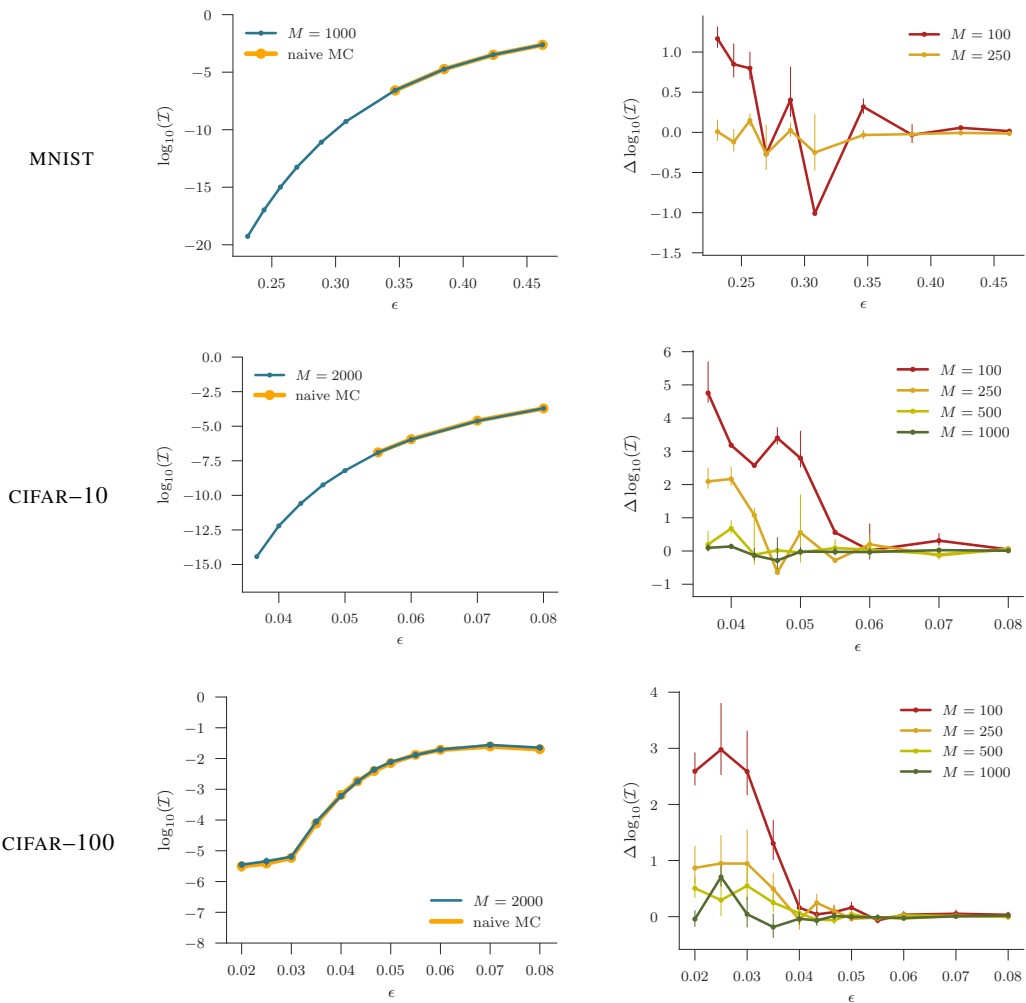

Figure 2: [Left] Estimates for $\mathcal{I}$ on adversarial properties of a single datapoint with $\rho = 0.1$, and $N \in \{10000, 10000, 300\}$ for MNIST/CIFAR−10/CIFAR−100 respectively. As in Figure 1, the error bars from 30 runs are barely visible, highlighting a very low variance in the estimates, while the close matching to the naive MC estimates when $\epsilon$ is large enough to make the latter viable, indicate a very low bias. For CIFAR−100 the error bars are shown for the naive estimates, as well, from 10 runs. [Right] The difference in the estimate for the other values of $M$ from $M \in \{1000, 2000, 2000\}$ for MNIST/CIFAR−10/CIFAR−100, respectively. The estimate steadily converges as $M$ increases, with larger $M$ more important for rarer events.

To demonstrate that our approach can be employed on large networks, we tested adversarial properties on the CIFAR−100 dataset and a much larger DenseNet architecture (Huang et al., 2017a), with depth and growth-rate 40 (approximately $2 \times 10^6$ parameters). Due to the larger model size, we set $N = 300$, the largest minibatch that could be held in memory (a larger $N$ could be used by looping over minibatches). The naive Monte Carlo estimates used $5 \times 10^6$ samples for about an hour of computation time per estimate, compared to between five to fifteen minutes for each AMLS estimate. The results are presented in the bottom row of Figure 2, showing that our algorithm agrees with the naive Monte Carlo estimate.

## 6.4 ROBUSTNESS OF PROVABLE DEFENSES DURING TRAINING

We now examine how our robustness metric varies for a ReLU network as that network is trained to be more robust against norm bounded perturbations to the inputs using the method of Wong & Kolter (2018). Roughly speaking, their method works by approximating the set of outputs resulting from perturbations to an input with a convex outer bound, and minimizing the worst case loss over this

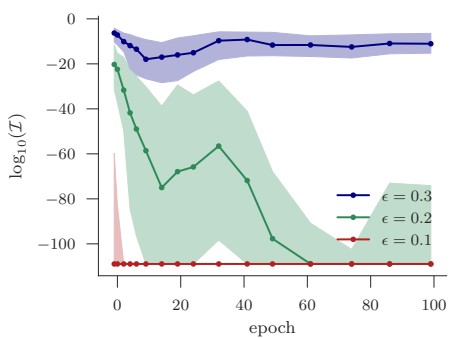

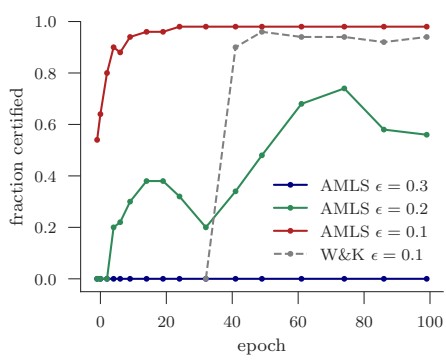

(a) Variation in $\mathcal{I}$ during robustness training

(b) Fraction of datapoints declared UNSAT

Figure 3: (a) Variation in $\mathcal{I}$ during the robustness training of on a CNN model for MNIST for three different perturbation sizes $\epsilon$. Epoch 0 corresponds to the network after conventional training, with further epochs corresponding to iterations of robustness training. The solid line indicates the median over 50 datapoints, and the limits of the shaded regions the 25 and 75 percentiles. Our measure is capped at $P_{\min} = \exp(-250)$. We see that while training improves robustness for $\epsilon = 0.2$, the initial network is already predominantly robust to perturbations of size $\epsilon = 0.1$, while the robustness to perturbations of size $\epsilon = 0.3$ actually starts to decrease after around 20 epochs. (b) Comparing the fraction of 50 datapoints for which Wong & Kolter (2018) produces a certificate-of-robustness for $\epsilon = 0.1$ ("W&K"), versus the fraction of those samples for which $\mathcal{I} = P_{\min}$ for $\epsilon \in \{0.1, 0.2, 0.3\}$ ("AMLS"). Due to very heavy memory requirements, it was computationally infeasible to calculate certificates-of-robustness for $\epsilon = \{0.2, 0.3\}$, and $\epsilon = 0.1$ before epoch 32 with the method of Wong & Kolter (2018). Our metric, however, suffers no such memory issues.

set. The motivation for this experiment is twofold. Firstly, this training provides a series of networks with ostensibly increasing robustness, allowing us to check if our approach produces robustness estimates consistent with this improvement. Secondly, it allows us to investigate whether the training to improve robustness for one type of adversarial attack helps to protect against others. Specifically, whether training for small perturbation sizes improves robustness to larger perturbations.

We train a CNN model on MNIST for 100 epochs with the standard cross-entropy loss, then train the network for a further 100 epochs using the robust loss of Wong & Kolter (2018), saving a snapshot of the model at each epoch. The architecture is the same as in (Wong & Kolter, 2018), containing two strided convolutional layers with 16 and 32 channels, followed by two fully connected layers with 100 and 10 hidden units, and ReLU activations throughout. The robustification phase trains the classifier to be robust in an $l_\infty$ $\epsilon$-ball around the inputs, where $\epsilon$ is annealed from 0.01 to 0.1 over the first 50 epochs. At a number of epochs during the robust training, we calculate our robustness metric with $\epsilon \in \{0.1, 0.2, 0.3\}$ on 50 samples from the test set. The results are summarized in Figure 3a with additional per-sample results in Appendix C.2. We see that our approach is able to capture variations in the robustnesses of the network.

As the method of Wong & Kolter (2018) returns the maximum value of the property for each sample over a convex outer bound on the perturbations, it is able to produce certificates-of-robustness for some datapoints. If the result returned is less than 0 then no adversarial examples exist in an $l_\infty$ ball of radius $\epsilon$ around that datapoint. If the result returned is greater than 0, then the datapoint may or may not be robust in that $l_\infty$ ball, due to fact that it optimizes over an outer bound.

Though we emphasize that the core aim of our approach is in providing richer information for SAT properties, this provides an opportunity to see how well it performs at establishing UNSAT properties relative to a more classical approach. To this end, we compared the fraction of the 50 samples from the test set that are verified by the method of Wong & Kolter (2018), to the fraction that have a negligible volume of adversarial examples, $\mathcal{I} = P_{\min}$, in their $l_\infty$ $\epsilon$-ball neighbourhood. The results are presented in Figure 3b.

Our method forms an upper bound on the fraction of robust samples, which can be made arbitrarily tighter by taking $P_{\min} \to 0$. Wong & Kolter (2018), on the other hand, forms a lower bound on the fraction of robust samples, where the tightness of the bound depends on the tightness of the convex outer bound, which is unknown and cannot be controlled. Though the true value must lie somewhere

between the two bounds, our bound still holds physical meaning it its own right in a way that Wong & Kolter (2018) does not: it is the proportion of samples for which the prevalence of violations is less than an a given acceptable threshold $P_{\min}$.

This experiment also highlights an important shortcoming of Wong & Kolter (2018). The memory usage of their procedure depends on how many ReLU activations cross their threshold over perturbations. This is high during initial training for $\epsilon = 0.1$ and indeed the reason why the training procedure starts from $\epsilon = 0.01$ and gradually anneals to $\epsilon = 0.1$. The result is that it is infeasible (the GPU memory is exhausted)—even for this relatively small model—to calculate the maximum value of the property on the convex outer bound for $\epsilon \in \{0.2, 0.3\}$ at all epochs, and $\epsilon = 0.1$ for epochs before 32. Even in this restricted setting where our metric has been reduced to a binary one, it appears to be more informative than that of Wong & Kolter (2018) for this reason.

## 7 DISCUSSION

We have introduced a new measure for the intrinsic robustness of a neural network, and have validated its utility on several datasets from the formal verification and deep learning literatures. Our approach was able to exactly emulate formal verification approaches for satisfiable properties and provide high confidence, accurate predictions for properties which were not. The two key advantages it provides over previous approaches are: a) providing an explicit and intuitive measure for *how* robust networks are to satisfiable properties; and b) providing improved scaling over classical approaches for identifying unsatisfiable properties.

Despite providing a more informative measure of how robust a neural network is, our approach may not be appropriate in all circumstances. In situations where there is an explicit and effective adversary, instead of inputs being generated by chance, we may care more about how far away the single closest counterexample is to the input, rather than the general prevalence of counterexamples. Here our method may fail to find counterexamples because they reside on a subset with probability less than $P_{\min}$; the counterexamples may even reside on a subset of the input space with measure zero with respect to the input distribution. On the other hand, there are many practical scenarios, such as those discussed in the introduction, where either it is unrealistic for there to be no counterexamples close to the input, the network (or input space) is too large to realistically permit formal verification, or where potential counterexamples are generated by chance rather than by an adversary. We believe that for these scenarios our approach offers significant advantages to formal verification approaches.

Going forward, one way the efficiency of our approach could be improved further is by using a more efficient base MCMC kernel in our AMLS estimator, that is, replace line 12 in Algorithm 1 with a more efficient base inference scheme. The current MH scheme was chosen on the basis of simplicity and the fact it already gave effective empirical performance. However, using more advanced inference approaches, such as gradient-based approaches like Langevin Monte Carlo (LMC) (Rossky et al., 1978) and Hamiltonian Monte Carlo (Neal, 2011), could provide significant speedups by improving the mixing of the Markov chains, thereby reducing the number of required MCMC transitions.

### ACKNOWLEDGMENTS

We gratefully acknowledge Sebastian Nowozin for suggesting to us to apply multilevel splitting to the problem of estimating neural network robustness. We also thank Rudy Bunel for his help with the COLLISIONDETECTION dataset, and Leonard Berrada for supplying a pretrained DenseNet model.

SW gratefully acknowledges support from the EPSRC AIMS CDT through grant EP/L015987/2. TR and YWT are supported in part by the European Research Council under the European Unions Seventh Framework Programme (FP7/20072013) / ERC grant agreement no. 617071. TR further acknowledges support of the ERC StG IDIU. MPK is supported by EPSRC grants EP/P020658/1 and TU/B/000048.

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

APPENDIX

## A METROPOLIS–HASTINGS

Metropolis–Hastings (MH) is an MCMC method that allows for sampling when one only has access to an unnormalized version of the target distribution (Gilks et al., 1995). At a high-level, one attempts iteratively proposes local moves from the current location of a sampler and then accepts or rejects this move based on the unnormalized density. Each iteration of this process is known as a MH transition.

The unnormalized targets distributions of interest for our problem are $\gamma_k(\mathbf{x})$ where

$$\pi_k(\mathbf{x}) \propto \gamma_k(\mathbf{x}) = p(\mathbf{x})\mathbb{1}_{\{s(\mathbf{x}) \geq L_k\}}, \quad k = 1, 2, \ldots, K.$$

A MH transition now consists of proposing a new sample using a proposal $\mathbf{x}' \sim g(\mathbf{x}' \mid \mathbf{x})$, where $\mathbf{x}$ indicates the current state of the sampler and $\mathbf{x}'$ the proposed state, calculating an acceptance probability,

$$A_k(\mathbf{x}' \mid \mathbf{x}) \triangleq \min\left\{1, \frac{\gamma_k(\mathbf{x}')g(\mathbf{x} \mid \mathbf{x}')}{\gamma_k(\mathbf{x})g(\mathbf{x}' \mid \mathbf{x})}\right\}, \quad (7)$$

and accepting the new sample with probability $A_k(\mathbf{x}' \mid \mathbf{x})$, returning the old sample if the new one is rejected. The proposal, $g(\mathbf{x}' \mid \mathbf{x})$, is a conditional distribution, such as a normal distribution centred at $\mathbf{x}$ with fixed covariance matrix. Successive applications of this transition process generates samples which converge in distribution to the target $\pi_k(\mathbf{x})$ and whose correlation with the starting sample diminishes to zero.

In our approach, these MH steps are applied independently to each sample in the set, while the only samples used for the AMLS algorithm are the final samples produced from the resulting Markov chains.

## B IMPLEMENTATION DETAILS

Algorithm 1 has computational cost $O(NMK)$, where the number of levels $K$ will depend on the rareness of the event, with more computation required for rarer ones. Parallelization over $N$ is possible provided that the batches fit into memory, whereas the loops over $M$ and $K$ must be performed sequentially.

One additional change we make from the approach outlined by Guyader et al. (2011) is that we perform MH updates on all chains in Lines 12, rather than only those that were previously killed off. This helps reduce the build up of correlations over multiple levels, improving performance.

Another is that we used an adaptive scheme for $g(\mathbf{x}'|\mathbf{x})$ to aid efficiency. Specifically, our proposal takes the form of a random walk, the radius of which, $\epsilon'$, is adapted to keep the acceptance ratio roughly around $0.234$ (see Roberts et al. (1997)). Each chain has a separate acceptance ratio that is average across MH steps, and after $M$ MH steps, for those chains whose acceptance ratio is below $0.234$ it is halved, and conversely for those above $0.234$, multiplied by $1.02$.

## C ADDITIONAL RESULTS

### C.1 VARYING $M$ FOR FIXED $\rho$ ON COLLISIONDETECTION

Whereas the exact value of $M$ within the range considered proved to not be especially important when $\rho = 0.1$, it transpires to have a large impact in the quality of the results for larger values of $\rho$ as shown in Figure 4.

### C.2 PER-SAMPLE ROBUSTNESS MEASURE DURING ROBUST TRAINING

Figure 5 illustrates the diverse forms that the per-sample robustness measure can take on the $40$ datapoints averaged over in Experiment §5.3. We see that different datapoints have quite varying initial levels of robustness, and that the training helps with some points more than others. In one case, the datapoint was still not robust add the end of training for the target perturbation size $\epsilon = 0.1$.

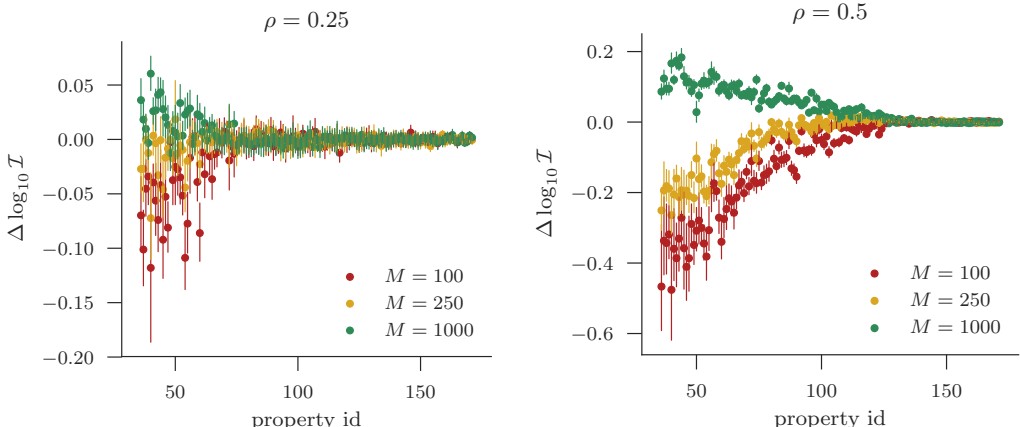

Figure 4: Mean AMLS estimate relative to naive (unbiased) MC estimate for different $M$ = holding $\rho$ fixed to $0.25$ (left) and $0.5$ (right), for those properties whose naive MC estimate was greater than $\log_{10} \mathcal{I} = -6.5$ such that they could be estimated accurately.

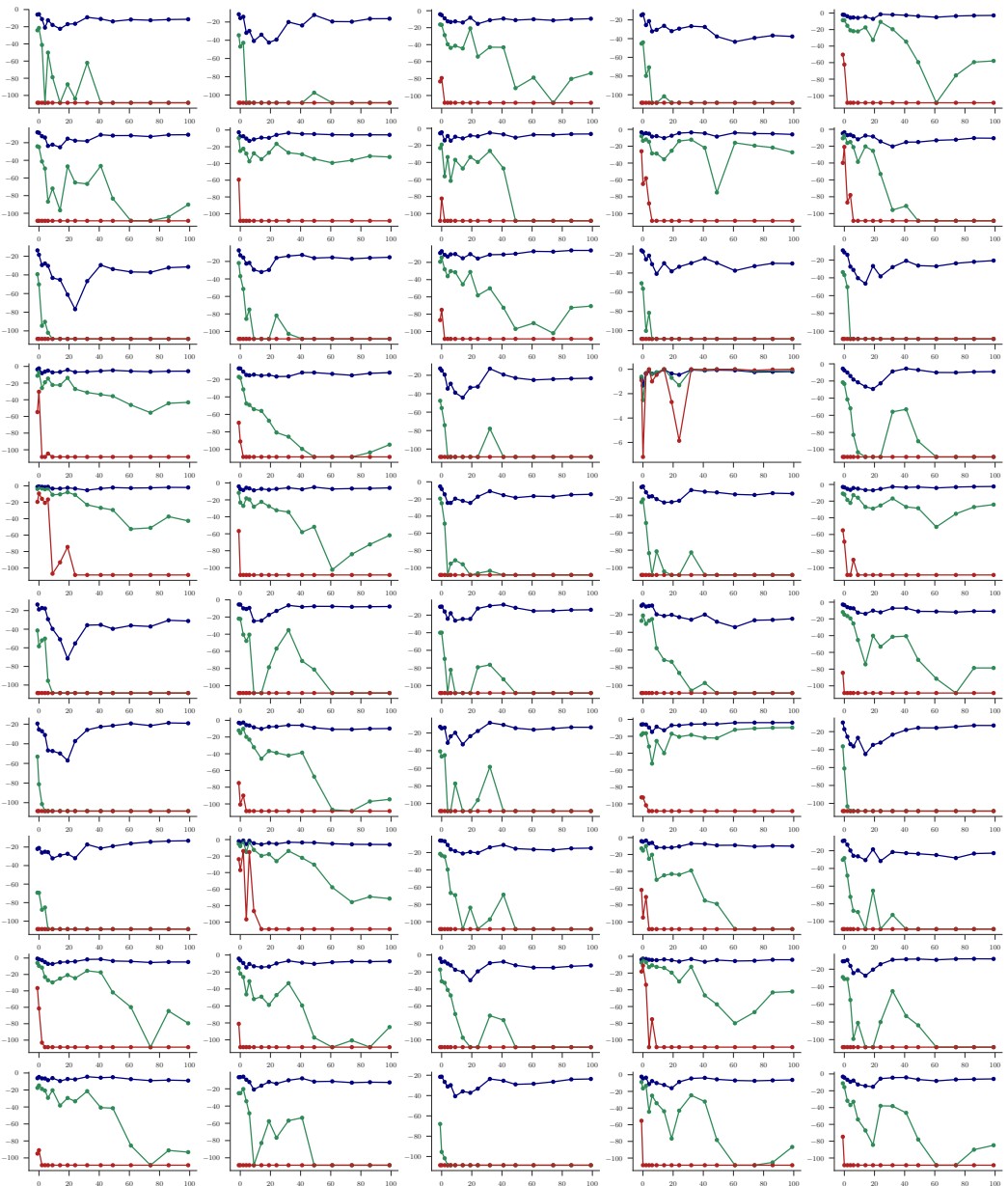

Figure 5: Convergence of individual datapoints used in forming Figure 3.

