# OpenReview forum: "A Statistical Approach to Assessing Neural Network Robustness"
_ICLR.cc/2019/Conference_

### Official Review · AnonReviewer1 · 2018-11-01
**Very interesting paper with a nice methodological tansfer between rare event estimation and NN verification**

**Rating:** 8
**Confidence:** 3

**Review:**

This is a paper of the verification of neural networks, i.e. check their robustness,
and the main contribution here is to tackle it as a statistical problem adressed with
multi-level splitting Monte Carlo approach. I found the paper well motivated and original,
resulting in a publishable piece of research up to a few necessary adjustments. These
concern principally notation issues and some potential improvements in the writing.
Let me list below some main remarks along the text, including also some typos.

* In the introduction, "the classical approach" is mentioned but to be the latter is
insufficiently covered. Some more detail would be welcome.

* page 2, "predict the probability": rather employ "estimate" in such context?

* "linear piecewise": "piecewise linear"?

* what is "an exact upper bound"?

* In related work, no reference to previous work on "statistical" approaches to NN
verification. Is it actually the case that this angle has never been explored so far?

* I am not an expert but to me "the density of adversarial examples" calls for further
explanation.

* From page 3 onwards: I was truly confused by the use of [x] throughought the text
(e.g. in Equation (4)). x is already present within the indicator, no need to add yet
another instance of it. Here and later I suffered from what seems to be like an awkward
attempts to stress dependency on variables that already appear or should otherwise
appear in a less convoluted way.

* In Section 4, it took me some time to understand that the considered metrics do not
require actual observations but rather concern coherence properties of the NN per se.
While this follows from the current framework, the paper might benefit from some more
explanation in words regarding this important aspect.

* In page 6, what is meant by "more perceptually similar to the datapoint"?

* In the discussion: is it really "a new measure" that is introduced here?

* In the appendix: the MH acronym should better be introduced, as should the notation
g(x,|x') if not done elsewhere (in which case a cross-reference would be welcome).
Besides this, writing "the last samples" requires disambiguation (using "respective"?).

---

> ### Author Response · Authors · 2018-11-17
> **Response to Reviewer #1**
>
> We are glad you found our work interesting and novel, and thank you for your helpful suggestions for improving the writing. We have taken them on board in the revised paper, making a number of edits.
>
> 1. "In the introduction, "the classical approach" is mentioned but to be the latter is
> insufficiently covered. Some more detail would be welcome."
>
> We have added a reference to what we mean by the classical approach in the related works section.
>
> 2. "page 2, "predict the probability": rather employ "estimate" in such context?"
>
> We have changed “predict” to “estimate”.
>
> 3. "'linear piecewise': 'piecewise linear'?"
>
> This was a typo and we have corrected this phrase.
>
> 4. "What is 'an exact upper bound'?"
>
> We mean that it is a true upper bound instead of just being a stochastic estimate of an upper bound (while, on the other hand, Weng et al’s approach is stochastic estimate of a lower bound).  However, we agree that the “exact” is superfluous and have removed it.
>
> 5. "I am not an expert but to me 'the density of adversarial examples' calls for further
> explanation."
>
> We think perhaps “the prevalence of adversarial examples” would be a better phrase and have corrected this. We mean that the input model density is integrated over for our metric to calculate the volume of counterexamples in a subset of the input domain, relative the overall volume of that input domain.
>
> 6. "From page 3 onwards: I was truly confused by the use of [x] throughout the text
> (e.g. in Equation (4)). x is already present within the indicator, no need to add yet
> another instance of it."
>
> In retrospect, we agree that this was confusing and have removed the [x] notation from the indicator function.
>
> 7. "In related work, no reference to previous work on "statistical" approaches to NN
> verification. Is it actually the case that this angle has never been explored so far?"
>
> As far as we are aware this is correct: we have not been able to find any prior work which aims to estimate the statistical prevalence of counterexamples.
>
> 8. "In page 6, what is meant by 'more perceptually similar to the datapoint'?"
>
> We mean that the minimal adversarial distortion for models on CIFAR-10 is known to typically be much smaller than for MNIST. The result of this is that an adversarial example on MNIST will often have visual salt-and-pepper noise, whereas an adversarial example for CIFAR-10 typically is indistinguishable to the naked eye from its unperturbed datapoint.
>
> 9. "In the appendix: the MH acronym should better be introduced, as should the notation
> g(x,|x') if not done elsewhere (in which case a cross-reference would be welcome).
> Besides this, writing "the last samples" requires disambiguation (using "respective"?)."
>
> We have added to this description so that it is less terse and more carefully introduces the notation, including changing “last samples” to “final samples” and adding in a reference for further reading.

---

### Official Review · AnonReviewer3 · 2018-11-03
**Interesting idea for quantitatively estimating the robustness of a network. Would like to see more comprehensive large-scale experiments.**

**Rating:** 7
**Confidence:** 4

**Review:**

Given a network and input model for generating adversarial examples, this paper presents an idea to quantitatively evaluate the robustness of the network to these adversarial perturbations. Although the idea is interesting, I would like to see more experimental results showing the scalability of the proposed method and for evaluating defense strategies against different types of adversarial attacks.  Detailed review below:
- How does the performance of the proposed method scale wrt scalability? It will be useful to do an ablation study, i.e. keep the input model fixed and slowly increase the dimension.
- Did you experiment with other MH proposal beyond a random walk proposal? Is it possible to measure the diversity of the samples using techniques such as the effective sample size (ESS) from the SMC literature?
- What is the performance of the proposed method against "universal adversarial examples"?
- The most interesting question is whether this method gives reasonable robustness estimates even for large networks such as AlexNet?
- Please provide some intuition for this line in Figure 3: "while the robustness to perturbations of size  = 0:3 actually starts to decrease after around 20 epochs."
- A number of attack and defense strategies have been proposed in the literature. Isn't it possible to use the proposed method to quantify the increase in the robustness towards an attack model using a particular defense strategy? If it is possible to show that the results of the proposed method match the conclusions from these papers, then this will be an important contribution.

---

> ### Author Response · Authors · 2018-11-17
> **Response to Reviewer #3**
>
> We thank you for your useful feedback and suggestions for additional experiments, and are glad you found the connection we draw between verification and rare event estimation to be an interesting idea.
>
> 1. "How does the performance of the proposed method scale wrt scalability? It will be useful to do an ablation study, i.e. keep the input model fixed and slowly increase the dimension."
>
> This is a great question and something we have been looking into. As a first step, we have run a new experiment at a higher scale with the CIFAR-100 dataset and a far larger DenseNet-40/40 architecture as discussed in the response to Reviewer 1.  We see our approach still performs very effectively on this larger problem, for which most existing verification approaches would struggle due to memory requirements (see also our new comparisons in Section 6.4).  We are now working on doing an ablation study on the size of the input dimension x, but it is unlikely we will be finished with this before the end of the rebuttal period due to the fact that it will require a very large number of runs to generate.
>
> 2. "Did you experiment with other MH proposal beyond a random walk proposal?"
>
> That’s an excellent idea and a topic for future research. We didn’t experiment with a MH proposal beyond a random walk because this was the simplest thing to try and it already worked well in practice.  As well as different proposals, we have also been thinking about the possibility to instead use a more advanced Langevin Monte Carlo approach to replace the MH, which we expect to mix more quickly as the chains are guided by the gradient information.
>
> 3. "What is the performance of the proposed method against 'universal adversarial examples'?"
>
> “Universal adversarial examples” refers to a method for constructing adversarial perturbations that generalize across data points for a given model, often generalizing across models too. Our method does not give a measure of robustness with respect to a particular attack method - it is attack agnostic. It measures in a sense the “volume” of adversarial examples around a given input, and so if this is negligible then the network is robustness to any attack for that subset of the input space, whether by a universal adversarial example or another method.  All the same, investigating the use of our approach in a more explicitly adversarial example setting presents an interesting opportunity for future work.
>
> 4. "The most interesting question is whether this method gives reasonable robustness estimates even for large networks such as AlexNet?"
>
> This is an important point to address.  As previously mentioned, we have extended the experiment of section 6.3 to use the much larger DenseNet-40/40 architecture on CIFAR-100 and we see that our method still performs admirably. See the updated paper and our response to Reviewer 1 above.
>
> 5. "Please provide some intuition for this line in Figure 3: 'while the robustness to perturbations of size epsilon=0.3 actually starts to decrease after around 20 epochs.'"
>
> The epsilon used during the training method of Wong and Kolter (ICML 2018) is annealed from 0.01 at epoch 0 to 0.1 at epoch 50. It’s interesting from Figure 5 that the network is made robust to epsilon = 0.1 and 0.2 by training to be robust using a much smaller epsilon. The network appears to become less robust for epsilon = 0.3 as the training epsilon reaches 0.1. So this a counterintuitive result that training using a smaller epsilon may be better for overall robustness. One hypothesis for this is that the convex outer adversarial polytope is insufficiently tight for larger epsilon. Another hypothesis may be that training with a lower epsilon has a greater effect on the adversarial gradient at an input, as the training happens on a perturbation closer to that input.
>
> 6. "A number of attack and defense strategies have been proposed in the literature. Isn't it possible to use the proposed method to quantify the increase in the robustness towards an attack model using a particular defense strategy? If it is possible to show that the results of the proposed method match the conclusions from these papers, then this will be an important contribution."
>
> It is possible to quantify the increase in robustness using a particular defense strategy, as we do in section 6.4 for the robust training method of Wong and Kolter (ICML 2018). We find that our method is in agreement with theirs. To quantify the increase in “robustness” with respect to a particular attack method, you can simply record the success of the attack method over samples from the test set as the training proceeds. This will not, however, be a reliable measure of robustness as the network can be trained to be resistant to the attack method in question while not being resistant to attack methods yet-to-be devised (the adversarial “arms race”). We believe that what we really desire is an attack agnostic robustness measure, such as the method in our work.

---

### Official Review · AnonReviewer2 · 2018-11-05
**Ok to accept after discussion**

**Rating:** 6
**Confidence:** 5

**Review:**

Verifying the properties of neural networks can be very difficult.  Instead of
finding a formal proof for a property that gives a True/False answer, this
paper proposes to take a sufficiently large number of samples around the input
point point and estimate the probability that a violation can be found.  Naive
Monte-Carlo (MC) sampling is not effective especially when the dimension is
high, so the author proposes to use adaptive multi-level splitting (AMLS) as a
sampling scheme. This is a good application of AMLS method.

Experiments show that AMLS can make a good estimate (similar quality as naive
MC with a large number of samples) while using much less samples than MC, on
both small and relatively larger models.  Additionally, the authors conduct
sensitivity analysis and run the proposed algorithm with many different
parameters (M, N, pho, etc), which is good to see.


I have some concerns on this paper:

I have doubts on applying the proposed method to higher dimensional inputs. In
section 6.3, the authors show an experiments in this case, but only on a dense
ReLU network with 2 hidden layers, and it is unknown if it works in general.
How does the number of required samples increases when the dimension of input
(x) increases?

Formally, if there exists a violation (counter-example) for a certain property,
and given a failure probability p, what is the upper bound of number of samples
(in terms of input dimension, and other factors) required so that the
probability we cannot detect this violation with probability less than p?
Without such a guarantee, the proposed method is not very useful because we
have no idea how confident the sampling based result is. Verification needs
something that is either deterministic, or a probabilistic result with a small
and bounded failure rate, otherwise it is not really a verification method.

The experiments of this paper lack comparisons to certified verification
methods. There are some scalable property verification methods that can give a
lower bound on the input perturbation (see [1][2][3]).  These methods can
guarantee that when epsilon is smaller than a threshold, no violations can be
found.  On the other hand, adversarial attacks give an upper bound of input
perturbation by providing a counter-example (violation). The authors should
compare the sampling based method with these lower and upper bounds. For
example, what is log(I) for epsilon larger than upper bound?

Additionally, in section 6.4, the results in Figure 2 also does not look very
positive - it unlikely to be true that an undefended network is predominantly
robust to perturbation of size epsilon = 0.1. Without any adversarial training,
adversarial examples (or counter-examples for property verification) with L_inf
distortion less than 0.1 (at least on some images) should be able to find. It
is better to conduct strong adversarial attacks after each epoch and see what
are the epsilons of adversarial examples.

Ideas on further improvement:

The proposed method can become more useful if it is not a point-wise method.
If given a point, current formal verification method can tell if a property is
hold or not.  However, most formal verification method cannot deal with a input
drawn from a distribution randomly (for example, an unseen test example). This
is the place where we really need a probabilistic verification method. The
setting in the current paper is not ideal because a probabilistic estimate of
violation of a single point is not very useful, especially without a guarantee
of failure rates.

For finding counter-examples for a property, using gradient based methods might
be a better way. The authors can consider adding Hamiltonian Monte Carlo to
this framework (See [4]).

References:
There are some papers from the same group of authors, and I merged them to one.
Some of these papers are very recent, and should be helpful for the authors
to further improve their work.

[1] "AI2: Safety and Robustness Certification of Neural Networks with Abstract
Interpretation", IEEE S&P 2018 by Timon Gehr, Matthew Mirman, Dana
Drachsler-Cohen, Petar Tsankov, Swarat Chaudhuri, Martin Vechev

(see also "Differentiable Abstract Interpretation for Provably Robust Neural
Networks", ICML 2018. by Matthew Mirman, Timon Gehr, Martin Vechev.  They also
have a new NIPS 2018 paper "Fast and Effective Robustness Certification" but is
not on arxiv yet)

[2] "Efficient Neural Network Robustness Certification with General Activation
Functions", NIPS 2018. by Huan Zhang, Tsui-Wei Weng, Pin-Yu Chen, Cho-Jui
Hsieh, Luca Daniel.

(see also "Towards Fast Computation of Certified Robustness for ReLU Networks",
ICML 2018 by Tsui-Wei Weng, Huan Zhang, Hongge Chen, Zhao Song, Cho-Jui Hsieh,
Duane Boning, Inderjit S. Dhillon, Luca Danie.)

[3] Provable defenses against adversarial examples via the convex outer
adversarial polytope, NIPS 2018. by Eric Wong, J. Zico Kolter.

(see also "Scaling provable adversarial defenses", NIPS 2018 by the same authors)

[4] "Stochastic gradient hamiltonian monte carlo." ICML 2014. by Tianqi Chen,
Emily Fox, and Carlos Guestrin.

============================================

After discussions with the authors, they agree to revise the paper according to our discussions and my primary concerns of this paper have been resolved. Thus I increased my rating.

---

> ### Author Response · Authors · 2018-11-17
> **Response to Reviewer #2 (2/2)**
>
> 4. "The experiments of this paper lack comparisons to certified verification
> methods. There are some scalable property verification methods that can give a
> lower bound on the input perturbation (see [1][2][3]).  These methods can
> guarantee that when epsilon is smaller than a threshold, no violations can be
> found.  On the other hand, adversarial attacks give an upper bound of input
> perturbation by providing a counter-example (violation). The authors should
> compare the sampling based method with these lower and upper bounds. For
> example, what is log(I) for epsilon larger than upper bound?"
>
> The three references and the follow-up work that you cite give different methods for obtaining a certificate-of-guarantee that a datapoint is robust in a fixed epsilon l_\infty ball, with varying levels of scalability/generality/ease-of-implementation. For those datapoints where they can produce such a certificate, the minimal adversarial distortion is lower-bounded by that fixed epsilon.
>
> This is important work to be sure, but we view it as predominantly orthogonal to ours, for which we define robustness differently, as the “volume” of adversarial examples rather than the distance to a single adversarial example. We actively argue that the minimal adversarial distortion is not a reliable measure of neural network robustness in many scenarios, as it is dictated by the position of a single violation, and conveys nothing about the amount of violations present.
>
> Despite these being two different definitions of robustness, to try and demonstrate some comparisons between the two, we extended experiment 6.4 (already using Wong and Kolter (ICML 2018) [3]) and compared the fraction of samples for which I = P_min to the fraction that could be certified by Wong and Kolter for epsilon in {0.1, 0.2, 0.3}. We found that it wasn’t possible to calculate the certificate of Wong and Kolter for epsilon = 0.2/0.3 for all epochs, or epsilon = 0.1 before a certain epoch, due to its exorbitant memory usage. This significant memory gain thus indicates that our approach may still have advantages when used as a method for approximately doing more classical verification, even though this was not our aim.  Please see the updated paper for full details.
>
> 5. "Additionally, in section 6.4, the results in Figure 2 also does not look very
> positive - it unlikely to be true that an undefended network is predominantly
> robust to perturbation of size epsilon = 0.1. Without any adversarial training,
> adversarial examples (or counter-examples for property verification) with L_inf
> distortion less than 0.1 (at least on some images) should be able to find."
>
> You are correct that without any robustness training it is possible to find adversarial examples with distortion less than 0.1 for some inputs.  This is indeed what our results show in Figure 5 in the appendices, illustrating our metric for individual samples. You can see for several samples that were not initially robustness to eps=0.1 perturbations (log(I) > log(P_min)), the value of log(I) decreases steadily as the robust training procedure is applied.
>
> It does appear, however, that the network is predominantly robust to perturbations smaller than 0.1 before robustness training. The curves in Figure 3 plot the values of our measure log(I) between the 25th and 75th percentile for a number of samples. This shows that the network is already robust to perturbations of size eps=0.1 for more than about 75% of samples before the training procedure of Kolter and Wong is applied.
>
> All the same, we agree that the original Figure 3 was confusing in this respect, and have rerun this experiment with a lower minimum threshold for log(I) to make the point clearer in the graph. With this lower value of log(P_min), we see the 75 percentile of log(I) over the samples quickly decrease as robustness training proceeds for eps=0.2. Notably, however, log(I) is incredibly small before any of this training for eps=0.1, demonstrating how it is important to not only think in terms of whether any violations are present, but also how many: here less the proportion of violating samples is less than 10^-100 at eps=0.1 for most of the datapoints.

---

> ### Author Response · Authors · 2018-11-17
> **Response to Reviewer #2 (1/2)**
>
> We thank Reviewer 1 for their critical appraisal and helpful suggestions.
>
> 1. "Instead of finding a formal proof for a property that gives a True/False answer, this
> paper proposes to take a sufficiently large number of samples around the input
> point point and estimate the probability that a violation can be found. "
>
> We would like to make it clear that our method is less about finding a probability that violation can be found and more about trying to provide more information that just this true/false answer.  In particular, establishing how prevalent violations are, rather than just the usual binary information that a single violation exists.  As such, our motivation is not to provide an approximation of classical formal verification methods, but to go beyond them and establish additional important information.  Note the important distinction here between “the probability of the event that the property is violated” (as per our abstract), which is to do with the proportion of samples which are violations, compared with “the probability that a violation can be found”, which is to do with the probability that any of the samples are violations.
>
> 2. "I have doubts on applying the proposed method to higher dimensional inputs. In
> section 6.3, the authors show an experiments in this case, but only on a dense
> ReLU network with 2 hidden layers, and it is unknown if it works in general.
> How does the number of required samples increases when the dimension of input
> (x) increases?"
>
> We agree this is an important point to address, and have extended section 6.3 to include an experiment with a DenseNet-40/40 architecture (with approx. 2 million parameters) for the CIFAR-100 dataset, producing a plot similar to Figure 2. The values agree with the naive (unbiased) Monte Carlo estimates where they can be feasibly calculated, similar to the existing results, thereby establishing the estimates still have low bias for this more difficult problem.  Furthermore, the variability in the results was very low (so much that it is not perceptible on the plot), thereby showing that the approach gives very low variance.  Together we believe this provides strong evidence that our approach is able to scale to large architectures.
>
> 3. "Formally, if there exists a violation (counter-example) for a certain property,
> and given a failure probability p, what is the upper bound of number of samples
> (in terms of input dimension, and other factors) required so that the
> probability we cannot detect this violation with probability less than p?
> Without such a guarantee, the proposed method is not very useful because we
> have no idea how confident the sampling based result is. Verification needs
> something that is either deterministic, or a probabilistic result with a small
> and bounded failure rate, otherwise it is not really a verification method."
>
> We want to stress that we are not claiming to perform formal verification or even an approximation of it. Namely, as alluded to before, we are not predicting a failure probability, but the prevalence of violations.  We believe this an advantage of the method as by relaxing the assumptions of formal verification, we are able to give not just a binary answer as to whether a neural network it is robust or not to a property, but a more informative quantitative measure telling how robust. We show empirically the bias/variance of our estimate is low in the experimental section. All the same, it is interesting to note that it should be possible in principle to derive the type of bounds that you speak of for UNSAT properties, by using appropriate learning theory techniques (e.g. https://arxiv.org/abs/1810.08240).  We think that this forms a very interesting direction for future work, but that it is beyond the scope of the current paper.

---

> > ### Comment · AnonReviewer2 · 2018-11-27
> > **Thank you for the response. I like this paper but unfortunately find its current representation misleading.**
> >
> > Thanks for clarifying that the paper is on finding "the probability of the event that the property is violated". In that interpretation, the results of this paper makes more sense. I also appreciate your effort on adding new experiments on more datasets. However, my major concerns still remain.
> >
> > Adversarial examples are about the worst case scenario, rather than the average case that can be represented by sampling. Many networks are pretty robust to very large Gaussian perturbations, but not robust to very tiny adversarial noise that is crafted using gradient ascent.
> >
> > More precisely, adversarial examples can live in a subspace which measures 0. For example, for a certain network with a 10-dimensional input (x_1, ..., x_10), all of its adversarial examples can be found only when x_1 = 0 (we can see x_1 as a "kill switch" of the network, when x_1 =/= 0  it behaves normally, and when x_1==0 it behaves badly). There are still infinite many adversarial examples in the hyperplane of x_1=0, and they can be arbitrarily close to normal examples (when x_1 has a small value). But using a sampling based approach, we can find these violations with a probability of 0 (as they measure 0).
> >
> > Thus, even if we cannot find any violations using a sampling based approach, we can hardly argue that the network illustrated above is robust. Especially, adversarial examples may lie in a low dimensional subspace, but the entire sampling space can be very high dimensional, so it is very inefficient to find violations in this way. Using AMLS might alleviate this issue, but cannot complete solve the problem.
> >
> > On the other hand, one benefit of sampling based method is that they can possibly scale to larger models/datasets. Current formal verification methods for neural networks only work on small networks.
> >
> > However, if we have to resort to sampling to find violations rather than using formal verification, there might be better and simpler ways that worth investigating more in this paper. For example, we can run simple Monte Carlo to sample K points, and run N steps of PGD on each points to find violations. Gradient based methods can possibly have a better chance on finding violations lying in a low dimensional subspace. So using MC based approach without gradient knowledge (or some form of Hamiltonian) might not be the most efficient way here.
> >
> > In my opinion, when using sampling based approaches for the application of verification, we should be very careful. For example, (Weng et al. 2018b) used sampling to estimate the robustness of neural networks; it is not even claimed to be a "verification" method, rather just an "estimation" of robustness. Even though, (Goodfellow, 2018) attacked it by showing that this sampling based method may fail silently (which is expected as this method does not have a guarantee), and advocate not using this method.
> >
> > Despite all my concern above, I actually like this paper because it is well motivated, includes extensive experimental results, and has a nice application of the AMLS method. If the paper were advertised differently (for example, as a method to "estimate" network robustness), I would recommend to accept this paper. But in this current form, I feel it misleading because "statistical verification" seems to be a new approach of formal verification, and gives people a strong feeling that it is capable to replacing other formal verification methods. Thus, I had to keep my original rating for now. I encourage the authors to rephrase the paper in a more conservative manner, investigate more on the HMC based sampling approach, and discuss the potential limitations and drawbacks when using sampling. This will become a very good paper.

---

> > > ### Author Response · Authors · 2018-11-27
> > > **Updates to ensure that the paper is not misconstrued**
> > >
> > > Thank you for your follow up comments, praise, and suggestions.  We are happy to take on board your constructive criticism and make edits to ensure that the paper is not misconstrued in any way. Unfortunately, the revision period for the paper ended last night so we are not able to update the submission itself, but we detail the changes we have already made locally to the paper in response to your suggestions below.  We hope these address your concerns and look forward to hearing your thoughts.
> > >
> > > 1) We have changed the title to "A Statistical Approach to Assessing Neural Network Robustness" and removed any references to "statistical verification" throughout the paper.  We have further made small edits throughout to ensure it is crystal clear we are not proposing a new approach for formal verification.
> > >
> > > 2) We have updated the abstract to the below to ensure there is no potential confusion.
> > >
> > > "We present a new approach to assessing the robustness of neural networks based on estimating the proportion of inputs for which a property is violated. Specifically, we estimate the probability of the event that the property is violated under an input model. Our approach critically varies from the formal verification framework in that when the property can be violated, it provides an informative notion of how robust the network is, rather than just the conventional assertion that the network is not verifiable. Furthermore, it provides an ability to scale to larger networks than formal verification approaches. Though the framework still provides a formal guarantee of satisfiability whenever it successfully finds one or more violations, these advantages do come at the cost of only providing a statistical estimate of unsatisfiability whenever no violation is found. Key to the practical success of our approach is an adaptation of multi-level splitting, a Monte Carlo approach for estimating the probability of rare events, to our statistical robustness framework. We demonstrate that our approach is able to emulate formal verification procedures on benchmark problems, while scaling to larger networks and providing reliable additional information in the form of accurate estimates of the violation probability."
> > >
> > > 3) We wholeheartedly agree that using gradient-based sampling methods in the place of the MH sampler could give noticeable empirical improvements -- this is something we were already looking into as a component of some follow-up work.  The MH sampler was chosen on the basis of simplicity and the fact it already gave sufficiently effective empirical performance.  We have added a paragraph to the discussion to highlight this point (see the updated discussion at the end of this reply).
> > >
> > > 4) We agree that further discussion on the relative advantages/disadvantages of the approach compared to formal verification, and, in particular, the respective scenarios where each is preferable, would strengthen the paper.  To this end, we have added a paragraph to the discussion on this point (see below).

---

> > > > ### Author Response · Authors · 2018-11-27
> > > > **Updated discussion section**
> > > >
> > > > “We have introduced a new measure for the intrinsic robustness of a neural network, and have validated its utility on several datasets from the formal verification and deep learning literatures. Our approach was able to exactly emulate formal verification approaches for satisfiable properties and provide high confidence, accurate predictions for properties which were not. The two key advantages it provides over previous approaches are: a) providing an explicit and intuitive measure for how robust networks are to satisfiable properties; and b) providing improved scaling over classical approaches for identifying unsatisfiable properties.
> > > >
> > > > Despite providing a more informative measure of how robust a neural network is, our approach may not be appropriate in all circumstances. In situations where there is an explicit and effective adversary, instead of inputs being generated by chance, we may care more about how far away the single closest counterexample is to the input, rather than the general prevalence of counterexamples. Here our method may fail to find counterexamples because they reside on a subset with probability less than Pmin; the counterexamples may even reside on a subset of the input space with measure zero with respect to the input distribution. On the other hand, there are many practical scenarios, such as those discussed in the introduction, where either it is unrealistic for there to be no counterexamples close to the input, the network (or input space) is too large to realistically permit formal verification, or where potential counterexamples are generated by chance rather than by an adversary. We believe that for these scenarios our approach offers significant advantages to formal verification approaches.
> > > >
> > > > Going forward, one way the efficiency of our approach could be improved further is by using a more efficient base MCMC kernel in our AMLS estimator, that is replace line 12 in Algorithm 1 with a more efficient base inference scheme. The current MH scheme was chosen on the basis of simplicity and the fact it already gave effective empirical performance. However, using more advanced inference approaches, such as gradient-based approaches like Langevin Monte Carlo (LMC) (Rossky et al., 1978) and Hamiltonian Monte Carlo (Neal, 2011), could provide significant speedups by improving the mixing of the Markov chains, thereby reducing the number of required MCMC transitions.”

---

> > > > > ### Comment · AnonReviewer2 · 2018-11-27
> > > > > **Thanks for the update. Most of my concerns are resolved.**
> > > > >
> > > > > Dear Paper 1101 Authors,
> > > > >
> > > > > Thanks for the clarification. The new abstract and title look much better than before, and the updated discussion section will greatly help readers understand the paper without misconstruction. It resolves most of my concerns.
> > > > >
> > > > > I will update the rating of this paper. Make sure to prepare the final revision of the paper based on the updates proposed above.
> > > > >
> > > > > Thanks,
> > > > > Paper1101 AnonReviewer2

---

### Author Response · Authors · 2018-11-17
**To Our Reviewers**

We would like to thank our reviewers for taking the time to read and evaluate our work and were glad to receive your detailed feedback, which we believe will improve the paper. To these ends, we have uploaded a revised version of the paper, with two additional experiments and a number of edits to address the concerns raised. In particular, we have added a new experiment with a substantially larger architecture to demonstrate the scaling of the approach, and adapted our final experiment to better demonstrate both the behavior of our approach and highlights the links and differences with classical verification approaches.

Please see our replies to each reviewer for our responses to individual points.

---

### Meta-Review · Area_Chair1 · 2018-12-15
**well-written paper addressing timely question**

**Confidence:** 4
**Recommendation:** Accept (Poster)

**Metareview:**

* Strengths

The paper addresses an important topic: how to bound the probability that a given “bad” event occurs for a neural network under some distribution of inputs. This could be relevant, for instance, in autonomous robotics settings where there is some environment model and we would like to bound the probability of an adverse outcome (e.g. for an autonomous aircraft, the time to crash under a given turbulence model). The desired failure probabilities are often low enough that direct Monte Carlo simulation is too expensive. The present work provides some preliminary but meaningful progress towards better methods of estimating such low-probability events, and provides some evidence that the methods can scale up to larger networks. It is well-written and of high technical quality.

* Weaknesses

In the initial submission, one reviewer was concerned that the term “verification” was misleading, as the methods had no formal guarantees that the estimated probability was correct. The authors proposed to revise the paper to remove reference to verification in the title and the text, and afterwards all reviewers agreed the work should be accepted. The paper also may slightly overstate the generality of the method. For instance, the claim that this can be used to show that adversarial examples do not exist is probably wrong---adversarial examples often occupy a negligibly small portion of the input space. There was also concern that most comparisons were limited to naive Monte Carlo.

* Discussion

While there was initial disagreement among reviewers, after the discussion all reviewers agree the paper should be accepted. However, we remind the authors to implement the changes promised during the discussion period.